# LLM-FE: Automated Feature Engineering for Tabular Data with LLMs as Evolutionary Optimizers

## Abstract

Automated feature engineering plays a critical role in improving predictive model performance for tabular learning tasks. Traditional automated feature engineering methods are limited by their reliance on pre-defined transformations within fixed, manually designed search spaces, often neglecting domain knowledge. Recent advances using Large Language Models (LLMs) have enabled the integration of domain knowledge into the feature engineering process. However, existing LLM-based approaches use direct prompting or rely solely on validation scores for feature selection, failing to leverage insights from prior feature discovery experiments or establish meaningful reasoning between feature generation and data-driven performance. To address these challenges, we propose LLM-FE, a novel framework that combines evolutionary search with the domain knowledge and reasoning capabilities of LLMs to automatically discover effective features for tabular learning tasks. LLM-FE formulates feature engineering as a program search problem, where LLMs propose new feature transformation programs iteratively, and data-driven feedback guides the search process. Our results demonstrate that LLM-FE consistently outperforms state-of-the-art baselines, showcasing generalizability across diverse models, tasks, and datasets.
Code: https://anonymous.4open.science/r/LLM-FE-5525

## 1 Introduction

Feature engineering, the process of transforming raw data into meaningful features for machine learning models, is crucial for improving predictive performance, particularly when working with tabular data Domingos (2012). In many tabular prediction tasks, well-designed features have been shown to significantly enhance the performance of tree-based models, often outperforming deep learning models that rely on learned representations Grinsztajn et al. (2022). However, data-centric tasks such as feature engineering are one of the most time-consuming and resource-intensive processes in the tabular learning workflow Anaconda; Hollmann et al. (2024), as they require experts and data scientists to explore many possible combinations in the vast combinatorial space of feature transformations. Classical feature engineering methods Kanter & Veeramachaneni (2015); Khurana et al. (2016; 2018); Horn et al. (2020); Zhang et al. (2023) construct extensive search spaces of feature processing operations, relying on various search and optimization techniques to identify the most effective features. However, these search spaces are mostly constrained by predefined, manually designed transformations and often fail to incorporate domain knowledge Zhang et al. (2023). Domain knowledge can serve as an invaluable prior for identifying these transformations, leading to reduced complexity and more interpretable and effective features Hollmann et al. (2024).

Recently, Large Language Models (LLMs) have emerged as a powerful solution to this challenge, offering access to extensive embedded domain knowledge that can be leveraged for feature engineering. While recent approaches have demonstrated promising results in incorporating this knowledge into automated feature discovery, current LLM-based methods Hollmann et al. (2024); Han et al. (2024) rely predominantly on direct prompting mechanisms or validation scores to guide the feature generation process. These approaches do not leverage insights from prior feature discovery experiments, thereby falling short of establishing meaningful reasoning between feature generation and data-driven performance.

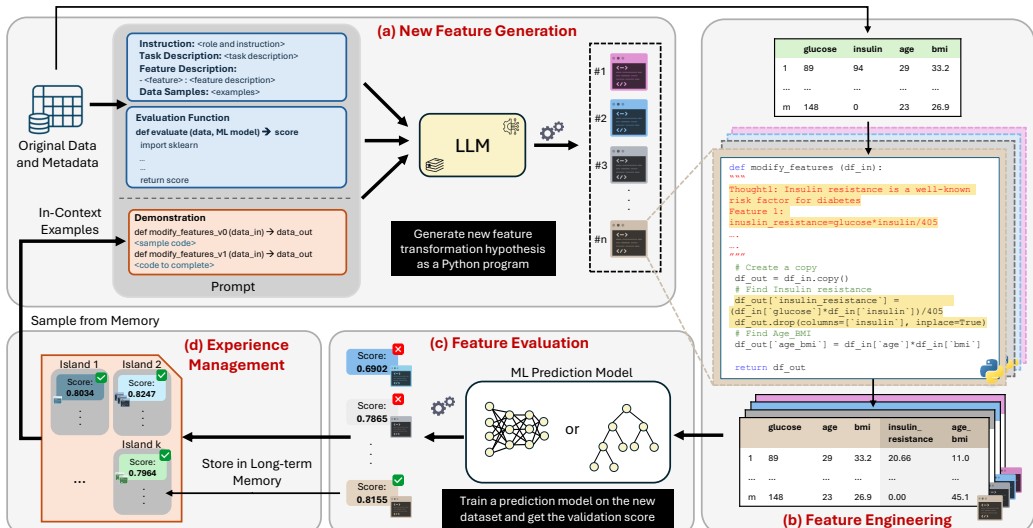

Figure 1: **Overview of the LLM-FE Framework.** For a given dataset, LLM-FE follows these steps: (a) **New Feature Generation**, where an LLM generates feature transformation hypotheses as programs for a given tabular dataset; (b) **Feature Engineering**, where the feature transformation program is applied to the underlying dataset, resulting in a modified dataset; (c) **Feature Evaluation**, where the modified dataset with the new features is evaluated using a prediction model; (d) **Experience Management**, which maintains a buffer of high-scoring programs that act as in-context samples for LLM's iterative refinement prompt. The features generated by LLM-FE are interpretable, using LLM's domain knowledge.

To address these limitations, we propose LLM-FE, *a novel framework integrating the capabilities of LLMs with tabular prediction models and evolutionary search to facilitate effective feature optimization.* As shown in Figure 1, LLM-FE follows an iterative process to generate and evaluate the hypothesis of the feature transformation, using the performance of the tabular prediction model as a reward to enhance the generation of effective features. Starting from an initial feature transformation program, LLM-FE leverages the LLMs' embedded domain knowledge by incorporating task-specific details, feature descriptions, and a subset of data samples to generate new feature discovery programs (Figure 1(a)). At each iteration, LLM acts as a knowledge-guided evolutionary optimizer, which mutates examples of previously successful feature transformation programs to generate new effective features Meyerson et al. (2024). The newly proposed features are then integrated with the original dataset to yield an augmented dataset (Figure 1(b)). The prediction model's performance is evaluated on a held-out validation set derived from the augmented dataset (Figure 1(c)), provides data-driven feedback that, combined with a dynamic memory of previously explored feature transformation programs (Figure 1(d)), guides the LLM to refine its feature generation iteratively.

Table 1 compares LLM-FE to several state-of-the-art classical and LLM-based feature engineering methods. Traditional methods lack adaptability and deeper contextual understanding, while LLM-based methods generate simple features Küken et al. (2024) or use feedback to iteratively refine only a single rule. In contrast, LLM-FE supports all four aspects by leveraging LLM-based domain knowledge and feedback-driven optimization to generalize

Table 1: Comparison of existing feature engineering methods.

| Method | Domain Knowledge | Feedback Driven | Complex Features | Multi-Feature Refinement |
|---|---|---|---|---|
| AutoFeat Horn et al. (2020) | ✗ | ✗ | ✓ | ✗ |
| OpenFE Zhang et al. (2023) | ✗ | ✗ | ✓ | ✗ |
| FeatLLM Han et al. (2024) | ✓ | ✗ | ✗ | ✗ |
| CAAFE Hollmann et al. (2024) | ✓ | ✓ | ✗ | ✗ |
| OCTree Nam et al. (2024) | ✓ | ✓ | ✗ | ✗ |
| **LLM-FE** | ✓ | ✓ | ✓ | ✓ |

well across table prediction tasks. We evaluate LLM-FE with `GPT-3.5-Turbo` OpenAI (2023) and `Llama-3.1-8B-Instruct` Dubey et al. (2024) backbones on classification and regression tasks across diverse tabular datasets. LLM-FE consistently outperforms the state-of-the-art feature engineering methods, identifying contextually relevant features that improve downstream performance. In particular, we observe improvements with tabular models like `XGBoost` Chen & Guestrin (2016),

`TabPFN` Hollmann et al. (2022), and `MLP` Gorishniy et al. (2021). Our analysis also highlights the importance of evolutionary search in achieving effective results. The major contributions of this work can be summarized as.

• We introduce LLM-FE, a novel framework that casts feature engineering as an LLM-guided evolutionary optimization problem, integrating domain knowledge, data-driven evaluation, and long-term memory for iterative refinement.

• Our experimental results demonstrate the effectiveness of LLM-FE, showcasing its ability to outperform state-of-the-art baselines, demonstrating generalizability across different predictors and LLM backbones.

• Through a comprehensive ablation study, we highlight the critical role of domain knowledge, evolutionary search, data-driven feedback, and data samples in guiding the LLM to efficiently explore the feature space and discover impactful features more effectively.

## 2 RELATED WORKS

**Feature Engineering.** Feature engineering involves creating meaningful features from raw data to improve predictive performance Hollmann et al. (2024). The growing complexity of datasets has driven the automation of feature engineering to reduce manual effort and optimize feature discovery. Traditional automated feature engineering methods include tree-based exploration, transformation enumeration, and learning-based methods Khurana et al. (2016); Kanter & Veeramachaneni (2015); Nargesian et al. (2017); Zhang et al. (2023). These traditional approaches often fail to leverage domain knowledge for feature discovery, making LLMs well-suited for such tabular prediction tasks due to their prior contextual domain understanding.

**LLMs and Optimization.** Advances in LLMs have shown that they can adapt to novel tasks via prompt engineering and in-context learning without retraining Brown et al. (2020); Wei et al. (2022). Yet, their outputs can be inconsistent or factually incorrect Madaan et al. (2024); Zhu et al. (2023), motivating research into mechanisms that refine or stabilize generations. A growing body of work has explored coupling LLMs with evaluators in iterative or evolutionary frameworks, where feedback, mutation, and crossover guide solution search Lehman et al. (2023); Wu et al. (2024); Meyerson et al. (2024). This paradigm has yielded progress in prompt optimization Yang et al. (2024); Guo et al. (2023), neural architecture search Zheng et al. (2023); Chen et al. (2024), mathematical heuristic discovery Romera-Paredes et al. (2024), and symbolic regression Shojaee et al. (2024). Building on this trajectory, our LLM-FE framework operationalizes LLMs as evolutionary optimizers, combining their rich prior knowledge with systematic, data-driven refinement to discover compact and high-performing features.

**LLMs for Tabular Learning.** The application of LLMs to structured data has typically relied on converting tables into textual representations Dinh et al. (2022); Hegselmann et al. (2023); Wang et al. (2023), or tailoring tokenization and pre-training strategies for tabular robustness Yan et al. (2024). For tabular prediction specifically, LLMs have been employed in fine-tuning and few-shot in-context paradigms Hegselmann et al. (2023); Nam et al. (2023), as well as in direct feature engineering. For example, FeatLLM Han et al. (2024) generates binary rules, while CAAFE Hollmann et al. (2024) exploits task descriptions to generate contextual features, and OCTree Nam et al. (2024) iteratively improves features through decision tree reasoning. However, these approaches often rely on incremental refinement of a single candidate. In contrast, LLM-FE maintains a diverse pool of promising programs and employs evolutionary search to efficiently traverse the feature space, leveraging mutation and crossover to uncover interpretable and data-driven transformations. This design enables the discovery of features that are not only predictive but also aligned with human interpretability, bridging the gap between domain-informed reasoning and optimization.

## 3 LLM-FE APPROACH

### 3.1 PROBLEM FORMULATION

A tabular dataset $\mathcal{D}$ comprises $N$ rows (or instances), each characterized by $d$ columns (or features). Each data instance $x_i$ is a $d$-dimensional feature vector with feature names denoted by $C = \{c_j\}_{j=1}^d$. The dataset is accompanied by metadata $\mathcal{M}$, which contains feature descriptions and task-specific information. For supervised learning tasks, each instance $x_i$ is associated with a corresponding label $y_i$, where $y_i \in \{0, 1, ..., K\}$ for classification tasks with $K$ classes, and $y_i \in \mathbb{R}$ for regression

tasks. Given a labeled tabular dataset $\mathcal{D} = (x_i, y_i)_{i=1}^N$ and prediction model $f$ to map from the input feature space $\mathcal{X}$ to its corresponding label space $\mathcal{Y}$, the feature engineering objective is to determine an optimal feature transformation $\mathcal{T}$, which enhances the performance of a predictive model when trained on the transformed input space. Formally, the feature engineering task can be defined as:

$$\max_{\mathcal{T}} \mathcal{E}(f^*(\mathcal{T}(\mathcal{X}_{\text{val}})), \mathcal{Y}_{\text{val}}) \tag{1}$$

subject to:

$$f^* = \arg\min_{f} \mathcal{L}_f(f(\mathcal{T}(\mathcal{X}_{\text{tr}})), \mathcal{Y}_{\text{tr}}) \tag{2}$$

where $(\mathcal{X}_{tr}, \mathcal{Y}_{tr})$ and $(\mathcal{X}_{val}, \mathcal{Y}_{val})$ are the sub-training set and validation set, respectively, that is derived from the training data $(\mathcal{X}_{train}, \mathcal{Y}_{train})$. The feature transformation $\mathcal{T}$ generated by the LLM $\pi_\theta$ and defined as $\mathcal{T} = \pi_\theta(\mathcal{X}_{train})$, meaning the transformation is learned from the training data by the LLM. The predictive model $f^*$ is then trained on the transformed training data $\mathcal{T}(\mathcal{X}_{train})$ to minimize loss. Consequently, the bilevel optimization problem seeks to identify the feature transformations $\mathcal{T}$ that maximize the performance $\mathcal{E}$ on $\mathcal{T}(\mathcal{X}_{\text{val}})$ while minimizing the loss function on the transformed training data, thereby efficiently exploring the potential feature space.

## 3.2 FEATURE GENERATION

Figure 1(a) illustrates the feature generation step that uses an LLM to create multiple new feature transformation programs, leveraging the model's prior knowledge, reasoning, and in-context learning abilities to effectively explore the feature space.

### 3.2.1 INPUT PROMPT

To facilitate the creation of effective and contextually relevant feature discovery programs, we develop a structured prompting methodology. The prompt is designed to provide comprehensive data-specific information, an initial feature transformation program for the evolution starting point, an evaluation function, and a well-defined output format (see Appendix D.2 for more details). Our input prompts $p$ are composed of the following key elements:

**Instruction.** The LLM is assigned the task of finding the most relevant features to help solve the given task. The task emphasizes using the LLM's prior knowledge of the dataset's domain to generate features. The LLM is explicitly instructed to generate novel features and provide clear step-by-step reasoning for their relevance to the prediction task. Moreover, since LLMs tend to generate simple features, we specifically instruct the LLM to generate complex features.

**Dataset Specification.** After providing the instructions, we provide LLM with the dataset-specific information from the metadata $\mathcal{M}$. This information encompasses a detailed description of the intended downstream task, along with the feature names $C$ and their corresponding descriptions. In addition, we provide a limited number of representative samples from the tabular dataset. To improve the effective interpretation of the data, we adopt the serialization approach used in previous works (Dinh et al., 2022; Hegselmann et al., 2023; Han et al., 2024). We serialized the data samples as follows:

$$\text{Serialize}(x_i, y_i, C) = \text{`If } c_1 \text{ is } x_i^1, ..., c_d \text{ is } x_i^d. \text{ Then Result is } y_i \text{'} \tag{3}$$

By providing dataset-specific details, we guide the language model to focus on the most contextually pertinent features that directly support the dataset and task objective.

**Evaluation Function.** The evaluation function, incorporated into the prompt, guides the language model to generate feature transformation programs that align with performance objectives. These programs augment the original dataset with new features, which are assessed on the basis of a prediction model's performance when trained on the augmented data. The model's evaluation score on the augmented validation set serves as an indicator of feature quality. By including the evaluation function in the prompt, the LLM generates programs that are inherently aligned with the desired performance criteria.

**In-Context Demonstration.** Specifically, we sample the $k$ highest-performing demonstrations from previous iterations, enabling the LLM to build on successful outputs. The iterative interaction between the LLM's generative outputs and the evaluator's feedback, informed by these examples, facilitates a systematic refinement process. With each iteration, the LLM progressively improves its outputs by leveraging patterns and insights identified in previous successful demonstrations.

### 3.2.2 FEATURE SAMPLING

At each iteration $t$, we construct the prompt $p_t$ by sampling the previous iteration as input to the LLM $\pi_\theta$, resulting in the output $\mathcal{T}_1, \ldots, \mathcal{T}_b = \pi_\theta(p_t)$ representing a set of $b$ sampled programs. To promote diversity and maintain a balance between exploration (creativity) and exploitation (prior knowledge), we employ stochastic temperature-based sampling. Each of the sampled feature transforms ($\mathcal{T}_i$) is executed before evaluation to discard error-prone programs. This ensures that only valid and executable feature transformation programs are considered further in the optimization pipeline. In addition, to ensure computational efficiency, a maximum execution time threshold is enforced, discarding any programs that exceed it.

### 3.3 DATA-DRIVEN EVALUATION

As illustrated in Figure 1(b), we use the generated features to augment the original dataset with the newly derived features. Similar to (Hollmann et al., 2024; Nam et al., 2024), our feature evaluation process comprises two stages: (i) model training on the augmented dataset, and (ii) performance assessment for feature quality (Figure 1(c)). We fit a tabular predictive model $f^*$, to the transformed training set $\mathcal{T}(\mathcal{X}_{\text{tr}})$, by minimizing the loss $\mathcal{L}_f$ as shown in Eq.1. Subsequently, we evaluated the LLM-generated feature transformations $\mathcal{T}$ by evaluating the model's performance on the augmented validation set $\mathcal{T}(\mathcal{X}_{\text{val}})$ (see Eqs. 1 and 2). As explained in Section 3.1, the objective is to find optimal features that maximize the performance $\mathcal{E}$, i.e., accuracy for classification and error metrics for regression.

### 3.4 EXPERIENCE MANAGEMENT

To promote diverse feature discovery and avoid stagnation in local optima, LLM-FE employs evolutionary multi-population experience management (Figure 1(d)) to store feature discovery programs in a dedicated database. Then, it uses samples from this database to construct in-context examples for LLM, facilitating the generation of novel features. This step consists of two components: (i) multi-population memory to maintain a long-term memory buffer, and (ii) sampling from this memory buffer to construct in-context example demonstrations. After evaluating the feature transforms in iteration $t$, we store the pair of feature transforms and score $(\mathcal{T}, s)$ in the population buffer $\mathcal{P}_t$ to iteratively refine the search process. To effectively evolve a population of programs, we adopt a multi-population model inspired by the 'island' model employed by (Cranmer, 2023; Shojaee et al., 2024; Romera-Paredes et al., 2024). The pro-

---

**Algorithm 1** LLM-FE

**Require:** LLM $\pi_\theta$, Dataset $\mathcal{D}$, Metadata $\mathcal{M}$, Iterations $T$, Model $f$, Metric $\mathcal{E}$
1: $\mathcal{P}_0 \leftarrow \texttt{BufferInit}()$
2: $\mathcal{T}^*, s^* \leftarrow \text{null}, -\infty$
3: $p \leftarrow \texttt{UpdatePrompt}(\mathcal{D}, \mathcal{M})$
4: **for** $t = 1$ **to** $T-1$ **do**
5:     $p_t \leftarrow p + \mathcal{P}_{t-1}.\texttt{topk}()$
6:     $\{\mathcal{T}_j\}_{j=1}^b \leftarrow \pi_\theta(p_t)$
7:     **for** $j = 1$ **to** $b$ **do**
8:         $s_j \leftarrow \texttt{FeatureScore}(f, \mathcal{T}_j, \mathcal{D}, \mathcal{E})$
9:         **if** $s_j > s^*$ **then**
10:             $\mathcal{T}^*, s^* \leftarrow \mathcal{T}_j, s_j$
11:         **end if**
12:         $\mathcal{P}_t \leftarrow \texttt{UpdateBuffer}(\mathcal{P}_{t-1}, \mathcal{T}_j, s_j)$
13:     **end for**
14: **end for**
15: **return** $\mathcal{T}^*, s^*$

---

gram population is divided into $m$ independent islands, each evolving separately and initialized with a copy of the user's initial example (see Figure 12(d)). This enables parallel exploration of the feature space, mitigating the risk of suboptimal solutions. At each iteration $t$, we select one of the $m$ islands and sample programs from the memory buffer to update the prompt with new in-context examples. The newly generated feature samples $b$ are evaluated, and if their scores $s_j$ exceed the current best score, the feature score pair $(\mathcal{T}_j, s_j)$ is added to the same island from which the in-context examples were sampled. To preserve diversity and ensure that programs with different performance characteristics are maintained in the buffer, we cluster programs within islands based on their signature, defined by their scores. To build refinement prompts, we follow the sampling process from (Romera-Paredes et al., 2024), first sampling one of the $m$ available islands, followed

by sampling the $k$ programs from the selected island to create $k$-shot in-context examples for the LLM. Cluster selection prefers high-scoring programs and follows Boltzmann sampling (De La Maza & Tidor, 1992) with a score-based probability of choosing a cluster $i$: $P_i = \frac{exp(s_i/\tau_c)}{\sum_i exp(s_i/\tau_c)}$, where $s_i$ denotes the mean score of the $i$-th cluster and $\tau_c$ is the temperature parameter. The sampled feature transformation programs from the memory buffer are then included in the prompt as examples to guide LLM toward successful feature transformations—incurring negligible computational overhead. Refer to Appendix B.4 for more details. Algorithm 1 presents the pseudocode of LLM-FE. We begin with the initialization of a memory buffer `BufferInit`, incorporating an initial population that contains a simple feature transform. This initialization serves as the starting point for the evolutionary search for feature transformation programs to be evolved in the subsequent steps. At each iteration $t$, the function `topk` is used to sample $k$ in-context examples from the population of the previous iteration $\mathcal{P}_{t-1}$ to update the prompt. Subsequently, we prompt the LLM using this updated prompt to sample $b$ new programs. The sampled programs are then evaluated using `FeatureScore`, which represents the Data-Driven Evaluation (Section 3.3). After $T$ iterations, the best-scoring program $\mathcal{T}^*$ from $\mathcal{P}_t$ and its score $s^*$ are returned as the optimal solution found for the problem. LLM-FE employs an iterative search to enhance programs, harnessing the LLM's capabilities. Learning from the evolving pool of experiences in its buffer, the LLM steers the search toward effective solutions.

## 4 EXPERIMENTAL SETUP

We evaluated LLM-FE on a range of tabular datasets, encompassing classification and regression tasks. Our experimental analysis included quantitative comparisons with baselines and detailed ablation studies. Specifically, we assessed our approach using three known tabular predictive models with distinct architectures: (1) `XGBoost`, a tree-based model Chen & Guestrin (2016), (2) `MLP`, a neural model Gorishniy et al. (2021), and (3) `TabPFN` Hollmann et al. (2022), a transformer-based foundation model Vaswani (2017). The results highlight LLM-FE's capability to generate effective features that consistently enhance the performance of different prediction models across datasets.

### 4.1 DATASETS

We followed Hollmann et al. (2024) to select datasets from previous feature engineering works like Han et al. (2024); Hollmann et al. (2024); Zhang et al. (2023) that include descriptive feature information. Our analysis contains 16 classification and 10 regression datasets, each containing mixed categorical and numerical features. We also include 8 large-scale, high-dimensional classification datasets to ensure comprehensive evaluation. These datasets were sourced from established machine learning repositories, including OpenML Vanschoren et al. (2014); Feurer et al. (2021), UCI Machine Learning Repository Asuncion et al. (2007), and Kaggle. Each dataset is accompanied by metadata, which includes a natural language description of the prediction task and descriptive feature names. We partitioned each dataset into train and test sets using an 80-20 split. Following Hollmann et al. (2024), we evaluated all methods over five iterations, each time using a distinct random seed and train-test splits. For more details, check Appendix C.

### 4.2 BASELINES

We evaluated LLM-FE against state-of-the-art feature engineering approaches, including OpenFE Zhang et al. (2023) and AutoFeat Horn et al. (2020), as well as LLM-based methods CAAFE Hollmann et al. (2024), FeatLLM Han et al. (2024) and OCTree Nam et al. (2024). We used `XGBoost` as the default tabular data prediction model in comparison with baselines and employed `GPT-3.5-Turbo` as the default LLM backbone for all LLM-based methods. To ensure a fair comparison, all LLM-based baselines were configured to query the LLM backbone for a total of 20 samples until they converged to their best performance. Appendix D.1 contains additional implementation details.

### 4.3 LLM-FE CONFIGURATION

In our experiments, we utilized `GPT-3.5-Turbo` and `Llama-3.1-8B-Instruct` as backbone LLMs, with a sampling temperature parameter of $t = 0.8$ and the number of islands set to $m = 3$. At each iteration, the LLM generated $b = 3$ feature transformation programs per prompt in Python. To ensure consistency with baselines, LLM-FE was also configured with a total of 20 LLM samples for each experiment. Finally, we sampled the top $m$ (where $m$ denotes the number of islands) feature discovery programs based on their respective validation scores and reported the final prediction through an ensemble. More implementation details are provided in Appendix D.2.

## 4.4 RESULTS AND DISCUSSION

In Table 2, we compare LLM-FE against various feature engineering baselines across 19 classification datasets. The results demonstrate that LLM-FE consistently enhances predictive performance from the base model (using raw data). LLM-FE also obtains the lowest mean rank (best performance) at a lower computational cost (see Appendix B.4), showing better effectiveness in enhancing feature discovery compared to other leading baselines. To further evaluate the effectiveness of LLM-FE, we perform experiments on 10 regression datasets using the same evaluation settings employed for the classification datasets. Due to the lack of regression data implementations in the available codebases for LLM-based baselines, in Table 3, we restrict our comparison to only non-LLM methods (OpenFE and AutoFeat), which have been previously validated on regression tasks. The results indicate that LLM-FE outperforms all baseline methods, achieving the lowest mean rank and consistently improving across all datasets. We provide additional analyses in Appendix A, including the effect of hyperparameter optimization on LLM-FE and evaluations with alternative predictive models such as CatBoost and Logistic Regression. We further study the transferability and generalizability of discovered features across different LLM backbones, showing that LLM-FE remains robust and effective under varied modeling and architectural choices.

Table 2: **Performance of XGBoost on Classification Datasets using various Feature Engineering (FE) Methods**, evaluated using accuracy (higher values indicate better performance). We report the mean values and standard deviation across five splits. ✗ : denotes execution time of greater than 12 hours or failure due to execution errors. **bold:** indicates the best performance. underline: indicates the second-best performance. 'n': indicates the number of samples; 'p': indicates the number of features.

| Dataset | n | p | Base | Classical FE Methods | | LLM-based FE Methods | | | LLM-FE |
|---|---|---|---|---|---|---|---|---|---|
| | | | | AutoFeat | OpenFE | CAAFE | FeatLLM | OCTree | |
| adult | 48.8k | 14 | $0.873 \pm 0.002$ | ✗ | $0.873 \pm 0.002$ | $0.872 \pm 0.002$ | $0.842 \pm 0.003$ | $0.870 \pm 0.002$ | $\mathbf{0.874} \pm \mathbf{0.003}$ |
| arrhythmia | 452 | 279 | $0.657 \pm 0.019$ | ✗ | ✗ | ✗ | ✗ | ✗ | $\mathbf{0.659} \pm \mathbf{0.018}$ |
| bank-marketing | 45.2k | 16 | $0.906 \pm 0.003$ | ✗ | $\mathbf{0.908} \pm \mathbf{0.002}$ | $0.907 \pm 0.002$ | $0.907 \pm 0.002$ | $0.900 \pm 0.002$ | $0.907 \pm 0.002$ |
| breast-w | 699 | 9 | $0.956 \pm 0.012$ | $0.956 \pm 0.019$ | $0.956 \pm 0.014$ | $0.960 \pm 0.009$ | $0.967 \pm 0.015$ | $0.969 \pm 0.009$ | $\mathbf{0.970} \pm \mathbf{0.009}$ |
| blood-transfusion | 748 | 4 | $0.742 \pm 0.012$ | $0.738 \pm 0.014$ | $0.747 \pm 0.025$ | $0.749 \pm 0.017$ | $\mathbf{0.771} \pm \mathbf{0.016}$ | $\underline{0.755} \pm 0.026$ | $0.751 \pm 0.036$ |
| car | 1728 | 6 | $0.995 \pm 0.003$ | $\underline{0.998} \pm 0.003$ | $0.998 \pm 0.003$ | $\mathbf{0.999} \pm \mathbf{0.001}$ | $0.808 \pm 0.037$ | $0.995 \pm 0.004$ | $\mathbf{0.999} \pm \mathbf{0.001}$ |
| cdc diabetes | 253k | 21 | $0.849 \pm 0.001$ | ✗ | $0.849 \pm 0.001$ | $0.849 \pm 0.001$ | $0.849 \pm 0.001$ | $0.849 \pm 0.001$ | $\mathbf{0.849} \pm \mathbf{0.001}$ |
| cmc | 1473 | 9 | $0.528 \pm 0.029$ | $0.505 \pm 0.015$ | $0.517 \pm 0.007$ | $0.524 \pm 0.016$ | $0.479 \pm 0.015$ | $0.525 \pm 0.027$ | $\mathbf{0.531} \pm \mathbf{0.015}$ |
| communities | 1.9k | 103 | $0.706 \pm 0.016$ | ✗ | $0.704 \pm 0.009$ | $0.707 \pm 0.013$ | $0.593 \pm 0.012$ | $0.708 \pm 0.016$ | $\mathbf{0.711} \pm \mathbf{0.012}$ |
| covtype | 581k | 54 | $0.870 \pm 0.001$ | ✗ | $\mathbf{0.885} \pm \mathbf{0.007}$ | $0.872 \pm 0.003$ | $0.554 \pm 0.001$ | $\underline{0.832} \pm 0.002$ | $0.882 \pm 0.003$ |
| credit-g | 1000 | 20 | $0.751 \pm 0.019$ | $0.757 \pm 0.017$ | $0.758 \pm 0.017$ | $0.751 \pm 0.020$ | $0.707 \pm 0.034$ | $0.753 \pm 0.021$ | $\mathbf{0.766} \pm \mathbf{0.015}$ |
| eucalyptus | 736 | 19 | $0.655 \pm 0.024$ | $0.664 \pm 0.028$ | $\underline{0.663} \pm 0.033$ | $\mathbf{0.679} \pm \mathbf{0.024}$ | ✗ | $0.658 \pm 0.041$ | $0.668 \pm 0.027$ |
| heart | 918 | 11 | $0.858 \pm 0.013$ | $0.857 \pm 0.021$ | $0.854 \pm 0.020$ | $0.849 \pm 0.023$ | $\underline{0.865} \pm 0.030$ | $0.852 \pm 0.022$ | $\mathbf{0.866} \pm \mathbf{0.021}$ |
| myocardial | 1.7k | 111 | $0.784 \pm 0.023$ | ✗ | $0.787 \pm 0.026$ | $\mathbf{0.789} \pm \mathbf{0.023}$ | $0.778 \pm 0.023$ | $0.787 \pm 0.031$ | $\mathbf{0.789} \pm \mathbf{0.023}$ |
| pc1 | 1109 | 21 | $0.931 \pm 0.004$ | $0.931 \pm 0.014$ | $0.931 \pm 0.009$ | $0.929 \pm 0.005$ | $0.933 \pm 0.007$ | $\underline{0.934} \pm 0.007$ | $\mathbf{0.935} \pm \mathbf{0.006}$ |
| vehicle | 846 | 18 | $0.754 \pm 0.016$ | $\mathbf{0.788} \pm \mathbf{0.018}$ | $\underline{0.785} \pm 0.008$ | $0.771 \pm 0.019$ | $0.744 \pm 0.035$ | $0.753 \pm 0.036$ | $0.761 \pm 0.027$ |
| **Mean Rank** | | | 4.26 | 4.89 | 3.26 | 3.31 | 4.94 | 3.84 | **1.47** |

Table 3: **Performance of XGBoost on Regression Datasets using various Feature Engineering (FE) Methods,** evaluated using normalized root mean square error (N-RMSE) (lower values indicate better performance). We report the mean and standard deviation across five splits. **bold:** indicates the best performance. underline: indicates the second-best performance. 'n': indicates the number of samples; 'p': indicates the number of features.

| Dataset | n | p | Base | Classical FE Methods | | LLM-FE |
|---|---|---|---|---|---|---|
| | | | | AutoFeat | OpenFE | |
| airfoil_self_noise | 1503 | 6 | $0.013 \pm 0.001$ | $\underline{0.012} \pm 0.001$ | $0.013 \pm 0.001$ | $\mathbf{0.011} \pm \mathbf{0.001}$ |
| bike | 17389 | 11 | $0.216 \pm 0.005$ | $\underline{0.223} \pm 0.006$ | $0.216 \pm 0.007$ | $\mathbf{0.207} \pm \mathbf{0.006}$ |
| cpu_small | 8192 | 10 | $0.034 \pm 0.003$ | $0.034 \pm 0.002$ | $0.034 \pm 0.002$ | $\mathbf{0.033} \pm \mathbf{0.003}$ |
| crab | 3893 | 8 | $0.234 \pm 0.009$ | $\underline{0.228} \pm 0.008$ | $0.224 \pm 0.001$ | $\mathbf{0.223} \pm \mathbf{0.013}$ |
| diamonds | 53940 | 9 | $0.139 \pm 0.002$ | $0.140 \pm 0.004$ | $\underline{0.137} \pm 0.002$ | $\mathbf{0.134} \pm \mathbf{0.002}$ |
| forest-fires | 517 | 13 | $1.469 \pm 0.080$ | $1.468 \pm 0.086$ | $\underline{1.448} \pm 0.113$ | $\mathbf{1.417} \pm \mathbf{0.083}$ |
| housing | 20640 | 9 | $0.234 \pm 0.009$ | $0.231 \pm 0.013$ | $\underline{0.224} \pm 0.005$ | $\mathbf{0.218} \pm \mathbf{0.009}$ |
| insurance | 1338 | 7 | $0.397 \pm 0.020$ | $0.384 \pm 0.024$ | $\underline{0.383} \pm 0.022$ | $\mathbf{0.381} \pm \mathbf{0.028}$ |
| plasma_retinol | 315 | 13 | $0.390 \pm 0.032$ | $0.411 \pm 0.036$ | $\underline{0.392} \pm 0.032$ | $\mathbf{0.388} \pm \mathbf{0.033}$ |
| wine | 4898 | 10 | $0.110 \pm 0.001$ | $0.109 \pm 0.001$ | $\underline{0.108} \pm 0.001$ | $\mathbf{0.105} \pm \mathbf{0.001}$ |
| **Mean Rank** | | | 3.40 | 3.10 | 2.20 | **1.00** |

## 5 ANALYSIS

### 5.1 GENERALIZABILITY ANALYSIS

To evaluate the generalizability of the LLM-FE, we examine its performance across multiple tabular prediction models and various LLM backbones. Specifically, we employ two LLM backbones, `Llama-3.1-8B-Instruct` and `GPT-3.5-Turbo`, in conjunction with three distinct tabular prediction models: `XGBoost` Chen & Guestrin (2016), a widely-used tree-based algorithm for tabular tasks; Multilayer Perceptron (`MLP`), a simple yet common deep-learning architecture tailored to tabular datasets Gorishniy et al. (2021); and `TabPFN` Hollmann et al. (2022), a recent transformer-based foundation model specifically designed for tabular data. Table 4 summarizes our findings, demonstrating that LLM-FE effectively identifies features that enhance the performance of various prediction models and LLM backbones across different tasks. Notably, the results indicate that features generated by LLM-FE using either LLM backbone consistently improve base model prediction performance compared to scenarios without any feature engineering.

### 5.2 ABLATION STUDY

We perform an ablation study on the classification datasets (<10,000 samples) listed in Table 2 to assess the contribution of each component in LLM-FE. Figure 2 illustrates the impact of individual components on overall performance, using `XGBoost` and `GPT-3.5-Turbo`. We report the accuracy aggregated and normalized over all the datasets. In the **'w/o Domain Knowledge'** setting, dataset and task-specific details are removed from the prompt and feature names are anonymized with generic placeholders such as $C_1, C_2, \ldots, C_n$. In this way, we remove any semantic meaning that could provide contextual insights about the problem. Without domain knowledge, the performance significantly drops to 0.626, underscoring its critical role in generating meaningful features. The **'w/o Evolutionary Refinement'** setting) also leads to the greatest decline in performance (0.587), emphasizing the importance of iterative data-driven feedback in addition to domain knowledge for refining feature transforms. Lastly, the results show that **'w/o Data Examples'** variant leads to only a slight performance drop, as LLMs might struggle to comprehend the nuances and patterns within the data samples. LLM-FE benefits significantly from each component, leading to an improvement.

Table 4: **Performance improvement by LLM-FE using different prediction models and LLM backbones.** We report the aggregated values for accuracy on classification tasks and normalized root mean square error on regression tasks. All results represent the mean and standard deviation computed across five splits. **bold:** indicates the best performance. TabPFN[*] evaluations are conducted using only 10,000 samples due to its limited processing capacity.

| Method | LLM | Classification ↑ | Regression ↓ |
|---|---|---|---|
| | | XGBoost | |
| Base | – | $0.820 \pm 0.020$ | $0.324 \pm 0.016$ |
| LLM-FE | Llama 3.1-8B | $0.832 \pm 0.021$ | $0.310 \pm 0.022$ |
| | GPT-3.5 Turbo | $\mathbf{0.840} \pm \mathbf{0.022}$ | $\mathbf{0.306} \pm \mathbf{0.015}$ |
| | | MLP | |
| Base | – | $0.745 \pm 0.034$ | $0.871 \pm 0.027$ |
| LLM-FE | Llama 3.1-8B | $0.768 \pm 0.032$ | $0.794 \pm 0.016$ |
| | GPT-3.5 Turbo | $\mathbf{0.791} \pm \mathbf{0.029}$ | $\mathbf{0.631} \pm \mathbf{0.043}$ |
| | | TabPFN[*] | |
| Base | – | $0.852 \pm 0.028$ | $0.289 \pm 0.016$ |
| LLM-FE | Llama 3.1-8B | $0.856 \pm 0.017$ | $0.288 \pm 0.016$ |
| | GPT-3.5 Turbo | $\mathbf{0.863} \pm \mathbf{0.018}$ | $\mathbf{0.286} \pm \mathbf{0.015}$ |

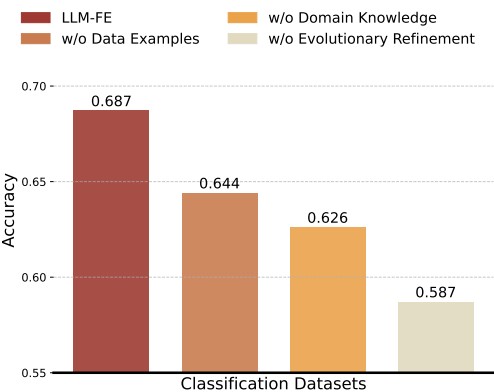

Figure 2: **Aggregated ablation study results across classification datasets**, showcasing the impact of individual components on LLM-FE's performance: (a) Data Examples, (b) Domain Knowledge, and (c) Evolutionary Refinement. Values are normalized with respect to the base LLM-FE model to facilitate fair comparison across conditions.

### 5.3 IMPACT OF DOMAIN KNOWLEDGE AND EVOLUTIONARY REFINEMENT

Figure 5 illustrates the qualitative benefits of incorporating domain knowledge into feature engineering. In this example, two approaches are contrasted: one without domain knowledge (Figure 5(a)), and

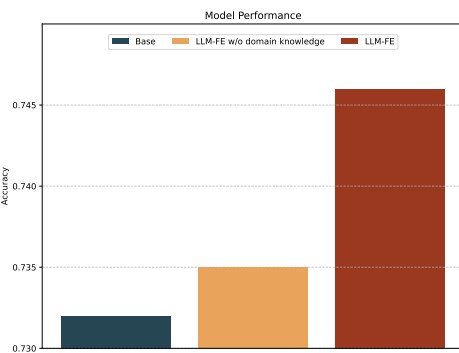

Figure 3: **Quantitative impact of domain knowledge on model accuracy.** Using domain knowledge boosts performance compared to both the base model and LLM-FE without domain knowledge.

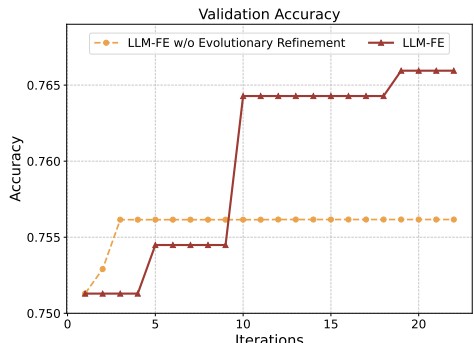

Figure 4: **Performance Trajectory Analysis.** for LLM-FE *w/o* evolutionary refinement and LLM-FE. LLM-FE demonstrates a better trajectory, highlighting the advantage of evolutionary refinement.

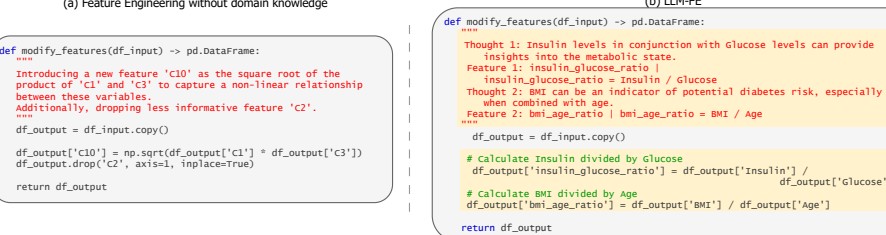

Figure 5: **Qualitative Analysis on Impact of Domain Knowledge.** illustrating how LLM-FE (b) utilizes domain knowledge to create meaningful features with descriptions , in contrast to feature engineering without domain insights (a) leading to uninterpretable outputs.

LLM-FE guided by domain-specific insights through an LLM-based feature engineering (Figure 5(b)). The domain-agnostic variant creates arbitrary transformations, such as combining features `C1` and `C3` using a square root of their product and dropping feature `C2` without clear justification. In contrast, LLM-FE leverages its embedded knowledge to derive interpretable and clinically meaningful features. Figure 3 presents a quantitative comparison of model performance on the same dataset, showing that LLM-FE with domain knowledge achieves the highest accuracy, outperforming both the base model and LLM-FE without domain knowledge. Figure 4 illustrates the validation accuracy trajectory of LLM-FE with and without evolutionary refinement across 20 iterations. The variant without refinement shows early improvement but quickly plateaus, indicating convergence to a local optimum. In contrast, LLM-FE continues to improve across iterations, achieving higher accuracy overall. This comparison highlights the effectiveness of evolutionary refinement in enhancing performance by enabling the model to escape local optima and optimize more effectively. Further analyses on feature interpretability, bias and memorization, and computational efficiency are provided in Appendix B.

## 6 CONCLUSION

In this work, we introduce a novel framework LLM-FE that leverages LLMs as evolutionary optimizers to discover new features for tabular prediction tasks. By combining LLM-driven hypothesis generation with data-driven feedback and evolutionary search, LLM-FE effectively automates the feature engineering process. Through comprehensive experiments on diverse tabular learning tasks, we demonstrate that LLM-FE consistently outperforms state-of-the-art baselines, delivering substantial improvements in predictive performance across various tabular prediction models. Future work could explore integrating more powerful or domain-specific language models to enhance the relevance and quality of generated features for domain-specific problems. Moreover, our framework could extend beyond feature engineering to other stages of the tabular learning and data-centric pipeline, such as data augmentation, automated data cleaning (including imputation and outlier detection), and model tuning.

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

# A ADDITIONAL RESULTS

## A.1 LLM-FE AND HYPERPARAMETER OPTIMIZATION (HPO)

To assess the impact of hyperparameter optimization (HPO) on LLM-FE, we conduct experiments with XGBoost and Multilayer Perceptron (MLP) models across five classification datasets where baseline models achieve accuracies below 0.8. We adopt the hyperparameter search spaces detailed in Table 6 (XGBoost) and Table 7 (MLP), following prior work Grinsztajn et al. (2022); Gorishniy et al. (2021). Optimization is performed with `Optuna` Akiba et al. (2019), using 400 trials with random sampling across multiple dataset splits. All MLP models are trained for up to 100 epochs with early stopping, retaining the checkpoint that achieves the best validation score. As summarized in Table 5, HPO consistently improves performance across all datasets for the Base model. Crucially, our proposed method LLM-FE delivers further gains even after HPO, highlighting that while HPO provides meaningful improvements, LLM-FE offers complementary and substantial enhancements that are independent of hyperparameter tuning.

Table 5: **Comparison of classification accuracy across datasets using Base and LLM-FE models**, evaluated under (a) without hyperparameter optimization (HPO) and (b) with HPO. Results are reported for both XGBoost and MLP.

| Dataset | XGBoost | | | | MLP | | | |
| --- | --- | --- | --- | --- | --- | --- | --- | --- |
| | *w/o* HPO | | *w/* HPO | | *w/o* HPO | | *w/* HPO | |
| | Base | LLM-FE | Base | LLM-FE | Base | LLM-FE | Base | LLM-FE |
| eucalyptus | $0.655 \pm 0.024$ | $\mathbf{0.668} \pm \mathbf{0.027}$ | $0.659 \pm 0.022$ | $\mathbf{0.678} \pm \mathbf{0.020}$ | $0.655 \pm 0.024$ | $\mathbf{0.668} \pm \mathbf{0.027}$ | $0.501 \pm 0.041$ | $\mathbf{0.506} \pm \mathbf{0.028}$ |
| credit-g | $0.751 \pm 0.019$ | $\mathbf{0.766} \pm \mathbf{0.025}$ | $0.761 \pm 0.022$ | $\mathbf{0.784} \pm \mathbf{0.017}$ | $0.558 \pm 0.144$ | $\mathbf{0.633} \pm \mathbf{0.101}$ | $0.689 \pm 0.032$ | $\mathbf{0.693} \pm \mathbf{0.028}$ |
| cmc | $0.528 \pm 0.030$ | $\mathbf{0.531} \pm \mathbf{0.015}$ | $0.554 \pm 0.026$ | $\mathbf{0.578} \pm \mathbf{0.021}$ | $0.559 \pm 0.020$ | $\mathbf{0.566} \pm \mathbf{0.020}$ | $\mathbf{0.572} \pm \mathbf{0.024}$ | $0.567 \pm 0.027$ |
| blood-transfusion | $0.674 \pm 0.017$ | $\mathbf{0.782} \pm \mathbf{0.017}$ | $0.777 \pm 0.021$ | $\mathbf{0.805} \pm \mathbf{0.009}$ | $0.674 \pm 0.071$ | $\mathbf{0.782} \pm \mathbf{0.017}$ | $0.616 \pm 0.182$ | $\mathbf{0.705} \pm \mathbf{0.078}$ |
| vehicle | $0.754 \pm 0.016$ | $\mathbf{0.761} \pm \mathbf{0.027}$ | $0.776 \pm 0.035$ | $\mathbf{0.801} \pm \mathbf{0.033}$ | $0.583 \pm 0.062$ | $\mathbf{0.673} \pm \mathbf{0.043}$ | $0.637 \pm 0.095$ | $\mathbf{0.694} \pm \mathbf{0.039}$ |

Table 6: XGBoost hyperparameters space.

| Parameter | Distribution |
|---|---|
| Max depth | UniformInt [1, 11] |
| Num estimators | UniformInt [100, 6100, 200] |
| Min child weight | LogUniformInt [1, 1e2] |
| Subsample | Uniform [0.5, 1] |
| Learning rate | LogUniform [1e-5, 0.7] |
| Col sample by level | Uniform [0.5, 1] |
| Col sample by tree | Uniform [0.5, 1] |
| Gamma | LogUniform [1e-8, 7] |
| Lambda | LogUniform [1, 4] |
| Alpha | LogUniform [1e-8, 1e2] |

Table 7: MLP hyperparameters space.

| Parameter | Distribution |
|---|---|
| Num layers | UniformInt [1, 8] |
| Layer size | UniformInt [16, 1024] |
| Dropout | Uniform [0, 0.5] |
| Learning rate | LogUniform [1e-5, 1e-2] |
| Category embedding size | UniformInt [64, 512] |
| Learning rate scheduler | {True, False} |
| Batch size | {256, 512, 1024} |

## A.2 TRANSFERABILITY OF GENERATED FEATURES

While traditional approaches typically use the same model for both feature generation and inference, we demonstrate that the features generated by one model can be utilized by other models. Following Nam et al. (2024), we use `XGBoost`, a computationally efficient decision tree-based model, to generate features to be used by more complex architectures for inference. As demonstrated in Table 8, `XGBoost`-generated features show an improvement in the performance of `MLP` and `TabPFN` over their base versions. This cross-architecture performance improvement suggests that the generated features capture meaningful data characteristics that are valuable across different modeling paradigms.

Table 8: **Comparative analysis of LLM-FE using feature transfer**. We use XGBoost to perform feature engineering and apply these features to MLP and TabPFN (indicated as LLM-FE $_{XGB}$). We report the accuracy for classification tasks and the normalized root mean square error for regression tasks. We report the mean and standard deviation across five random splits. **bold:** indicates the best performance.

| Method | LLM | Classification ↑ | Regression ↓ |
|---|---|---|---|
| MLP | | | |
| Base | – | $0.745 \pm 0.034$ | $0.871 \pm 0.027$ |
| **LLM-FE**$_{XGB}$ | GPT-3.5-Turbo | $0.763 \pm 0.030$ | $0.848 \pm 0.017$ |
| **LLM-FE** | GPT-3.5-Turbo | $\mathbf{0.791 \pm 0.029}$ | $\mathbf{0.631 \pm 0.043}$ |
| TabPFN | | | |
| Base | – | $0.852 \pm 0.028$ | $0.289 \pm 0.016$ |
| **LLM-FE**$_{XGB}$ | GPT-3.5-Turbo | $0.861 \pm 0.017$ | $0.287 \pm 0.015$ |
| **LLM-FE** | GPT-3.5-Turbo | $\mathbf{0.863 \pm 0.018}$ | $\mathbf{0.286 \pm 0.015}$ |

## A.3 ADDITIONAL MODELS

We extend the results from Section 4.4, showcasing the performance improvements achieved by LLM-FE across various prediction models. Specifically, we employ `XGBoost`, `MLP`, and `TabPFN` to generate features and subsequently use the same models for inference. As shown in Table 9, the features using `GPT-3.5-Turbo` by LLM-FE consistently enhance model performance across different datasets, outperforming the base versions trained without feature engineering. To further assess the generalizability of LLM-FE, we conducted experiments on smaller prediction models like CatBoost and Logistic Regression. From Table 10 that LLM-FE outperforms the respective base models for most of the datasets.

Table 9: **Performance improvement with LLM-FE.** We report the mean and standard deviation over five splits. We use Normalized Root Mean Square Error for all regression datasets, with a lower value indicating better performance, and Accuracy for classification datasets, with a higher value indicating better performance. **bold:** indicates the best performance.

| Dataset | XGBoost | | MLP | | TabPFN | |
|---|---|---|---|---|---|---|
| | Base | LLM-FE | Base | LLM-FE | Base | LLM-FE |
| Classification Datasets | | | | | | |
| breast-w | $0.956 \pm 0.012$ | $\mathbf{0.970} \pm \mathbf{0.009}$ | $0.957 \pm 0.010$ | $\mathbf{0.964} \pm \mathbf{0.005}$ | $\mathbf{0.971} \pm \mathbf{0.006}$ | $\mathbf{0.971} \pm \mathbf{0.007}$ |
| blood-transfusion | $0.742 \pm 0.012$ | $\mathbf{0.751} \pm \mathbf{0.036}$ | $0.674 \pm 0.071$ | $\mathbf{0.782} \pm \mathbf{0.017}$ | $0.790 \pm 0.012$ | $\mathbf{0.791} \pm \mathbf{0.011}$ |
| car | $0.995 \pm 0.003$ | $\mathbf{0.999} \pm \mathbf{0.001}$ | $0.929 \pm 0.019$ | $\mathbf{0.950} \pm \mathbf{0.009}$ | $0.984 \pm 0.007$ | $\mathbf{0.996} \pm \mathbf{0.006}$ |
| cmc | $0.528 \pm 0.030$ | $\mathbf{0.531} \pm \mathbf{0.015}$ | $0.559 \pm 0.028$ | $\mathbf{0.566} \pm \mathbf{0.028}$ | $0.563 \pm 0.030$ | $\mathbf{0.566} \pm \mathbf{0.036}$ |
| credit-g | $0.751 \pm 0.019$ | $\mathbf{0.766} \pm \mathbf{0.025}$ | $0.558 \pm 0.144$ | $\mathbf{0.633} \pm \mathbf{0.101}$ | $0.728 \pm 0.008$ | $\mathbf{0.794} \pm \mathbf{0.022}$ |
| eucalyptus | $0.655 \pm 0.024$ | $\mathbf{0.668} \pm \mathbf{0.027}$ | $0.414 \pm 0.064$ | $\mathbf{0.456} \pm \mathbf{0.062}$ | $0.712 \pm 0.016$ | $\mathbf{0.715} \pm \mathbf{0.021}$ |
| heart | $0.858 \pm 0.013$ | $\mathbf{0.866} \pm \mathbf{0.021}$ | $0.840 \pm 0.010$ | $\mathbf{0.844} \pm \mathbf{0.006}$ | $\mathbf{0.882} \pm \mathbf{0.025}$ | $0.880 \pm 0.021$ |
| pc1 | $0.931 \pm 0.004$ | $\mathbf{0.935} \pm \mathbf{0.006}$ | $\mathbf{0.931} \pm \mathbf{0.002}$ | $0.904 \pm 0.055$ | $0.936 \pm 0.007$ | $\mathbf{0.937} \pm \mathbf{0.003}$ |
| vehicle | $0.754 \pm 0.016$ | $\mathbf{0.761} \pm \mathbf{0.027}$ | $0.583 \pm 0.062$ | $\mathbf{0.673} \pm \mathbf{0.043}$ | $0.852 \pm 0.016$ | $\mathbf{0.856} \pm \mathbf{0.028}$ |
| Regression Datasets | | | | | | |
| airfoil_self_noise | $0.013 \pm 0.001$ | $\mathbf{0.011} \pm \mathbf{0.001}$ | $0.275 \pm 0.008$ | $\mathbf{0.108} \pm \mathbf{0.001}$ | $0.008 \pm 0.001$ | $\mathbf{0.007} \pm \mathbf{0.001}$ |
| bike | $0.216 \pm 0.005$ | $\mathbf{0.207} \pm \mathbf{0.005}$ | $0.636 \pm 0.015$ | $\mathbf{0.551} \pm \mathbf{0.022}$ | $0.200 \pm 0.005$ | $\mathbf{0.199} \pm \mathbf{0.006}$ |
| cpu_small | $0.034 \pm 0.003$ | $\mathbf{0.033} \pm \mathbf{0.003}$ | $3.793 \pm 0.731$ | $\mathbf{2.360} \pm \mathbf{1.263}$ | $0.036 \pm 0.001$ | $\mathbf{0.035} \pm \mathbf{0.001}$ |
| crab | $0.234 \pm 0.009$ | $\mathbf{0.223} \pm \mathbf{0.014}$ | $0.214 \pm 0.010$ | $\mathbf{0.212} \pm \mathbf{0.011}$ | $0.208 \pm 0.013$ | $\mathbf{0.207} \pm \mathbf{0.014}$ |
| diamond | $0.139 \pm 0.002$ | $\mathbf{0.134} \pm \mathbf{0.002}$ | $0.296 \pm 0.018$ | $\mathbf{0.265} \pm \mathbf{0.011}$ | $0.132 \pm 0.005$ | $\mathbf{0.130} \pm \mathbf{0.005}$ |
| forest-fires | $1.469 \pm 0.080$ | $\mathbf{1.417} \pm \mathbf{0.083}$ | $1.423 \pm 0.104$ | $\mathbf{1.344} \pm \mathbf{0.091}$ | $1.270 \pm 0.101$ | $\mathbf{1.269} \pm \mathbf{0.114}$ |
| housing | $0.234 \pm 0.009$ | $\mathbf{0.218} \pm \mathbf{0.009}$ | $0.505 \pm 0.009$ | $\mathbf{0.444} \pm \mathbf{0.036}$ | $0.210 \pm 0.004$ | $\mathbf{0.202} \pm \mathbf{0.003}$ |
| insurance | $0.397 \pm 0.144$ | $\mathbf{0.381} \pm \mathbf{0.142}$ | $0.896 \pm 0.053$ | $\mathbf{0.487} \pm \mathbf{0.026}$ | $0.351 \pm 0.018$ | $\mathbf{0.346} \pm \mathbf{0.020}$ |
| plasma_retinol | $0.390 \pm 0.032$ | $\mathbf{0.388} \pm \mathbf{0.033}$ | $0.440 \pm 0.070$ | $\mathbf{0.411} \pm \mathbf{0.053}$ | $\mathbf{0.348} \pm \mathbf{0.048}$ | $\mathbf{0.348} \pm \mathbf{0.055}$ |
| wine | $0.110 \pm 0.001$ | $\mathbf{0.105} \pm \mathbf{0.001}$ | $\mathbf{0.125} \pm \mathbf{0.001}$ | $\mathbf{0.125} \pm \mathbf{0.013}$ | $0.117 \pm 0.004$ | $\mathbf{0.116} \pm \mathbf{0.004}$ |

Table 10: **Performance improvement with LLM-FE on CatBoost and Logistic Regression.** We report the mean and standard deviation over five splits. We use Accuracy for classification datasets, with a higher value indicating better performance. **bold:** indicates the best performance.

| Dataset | Logistic Regression | | CatBoost | |
|---|---|---|---|---|
| | Base | LLM-FE | Base | LLM-FE |
| breast-w | $0.955 \pm 0.014$ | $\mathbf{0.962} \pm \mathbf{0.008}$ | $0.957 \pm 0.009$ | $\mathbf{0.962} \pm \mathbf{0.008}$ |
| blood-transfusion | $\mathbf{0.799} \pm \mathbf{0.014}$ | $\mathbf{0.799} \pm \mathbf{0.009}$ | $0.742 \pm 0.012$ | $\mathbf{0.751} \pm \mathbf{0.036}$ |
| car | $0.690 \pm 0.017$ | $\mathbf{0.696} \pm \mathbf{0.031}$ | $\mathbf{0.999} \pm \mathbf{0.001}$ | $\mathbf{0.999} \pm \mathbf{0.001}$ |
| cmc | $0.520 \pm 0.019$ | $\mathbf{0.525} \pm \mathbf{0.012}$ | $0.518 \pm 0.028$ | $\mathbf{0.548} \pm \mathbf{0.027}$ |
| credit-g | $0.764 \pm 0.006$ | $\mathbf{0.780} \pm \mathbf{0.015}$ | $\mathbf{0.714} \pm \mathbf{0.046}$ | $0.700 \pm 0.021$ |
| eucalyptus | $\mathbf{0.671} \pm \mathbf{0.036}$ | $0.667 \pm 0.042$ | $0.436 \pm 0.027$ | $\mathbf{0.509} \pm \mathbf{0.050}$ |
| heart | $\mathbf{0.877} \pm \mathbf{0.021}$ | $0.872 \pm 0.025$ | $\mathbf{0.845} \pm \mathbf{0.015}$ | $0.839 \pm 0.018$ |
| pc1 | $0.931 \pm 0.003$ | $\mathbf{0.935} \pm \mathbf{0.003}$ | $0.929 \pm 0.005$ | $\mathbf{0.932} \pm \mathbf{0.012}$ |
| vehicle | $\mathbf{0.772} \pm \mathbf{0.028}$ | $0.769 \pm 0.015$ | $0.719 \pm 0.045$ | $\mathbf{0.725} \pm \mathbf{0.033}$ |

## A.4 ROBUSTNESS TO NOISE

Noise is an inherent challenge in real-world data, arising from various sources, including sensor errors, human mistakes, environmental factors, and equipment limitations. Such noise can mask underlying patterns and impair machine learning models' ability to learn true relationships in the data. To evaluate how effectively LLM-FE leverages prior knowledge and evolutionary search to handle noisy data, we introduced Gaussian noise ($\sigma = 0, 0.01, 0.05, 0.1$) into numerical classification datasets. As shown in Figure 6, we compared `XGBoost`'s performance across different feature engineering approaches, using `GPT-3.5-Turbo` as the LLM backbone for both the LLM-based approaches. The results demonstrate that LLM-FE maintains superior accuracy and robustness even under increasing noise conditions.

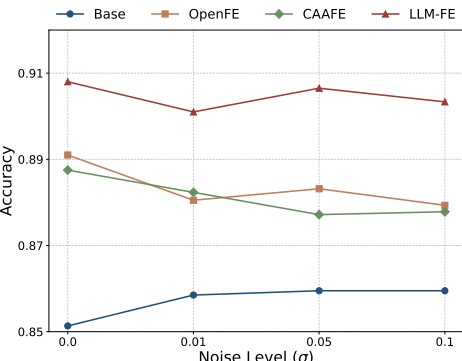

Figure 6: **Impact of Noise Levels** on XGBoost model performance across different feature engineering approaches, under increasing noise conditions ($\sigma = 0.0$ to $0.1$). We report the mean accuracy across six classification datasets containing only numerical features.

# B QUALITATIVE ANALYSIS

## B.1 INTERPRETABILITY ANALYSIS

As illustrated in Figure 7, LLM-FE generates feature-transformation programs in natural language, thus supporting interpretability. Each generated feature program is evaluated independently, and successful ones are stored for evolutionary refinement, enabling early discoveries to compose into higher-order features while preserving interpretability. To evaluate the utility of the generated features, we conduct attribution analysis using SHAP values. The results demonstrate that a consistent subset of discovered features receives high attribution scores, indicating that they actively contribute to the prediction process rather than serving as spurious or unused augmentations. Specifically, 16.7% of generated features rank among the top-10 most impactful features, and over 60% appear within the top-50 (Table 11), providing strong evidence that the features discovered by LLM-FE meaningfully enhance model performance and decision-making.

Table 11: Percentage of generated features ranked among the top-$k$ most impactful features by SHAP.

| Top-$k$ | Percentage |
|---------|------------|
| Top-10  | 16.67      |
| Top-20  | 25.93      |
| Top-30  | 37.04      |
| Top-40  | 57.41      |
| Top-50  | 62.96      |

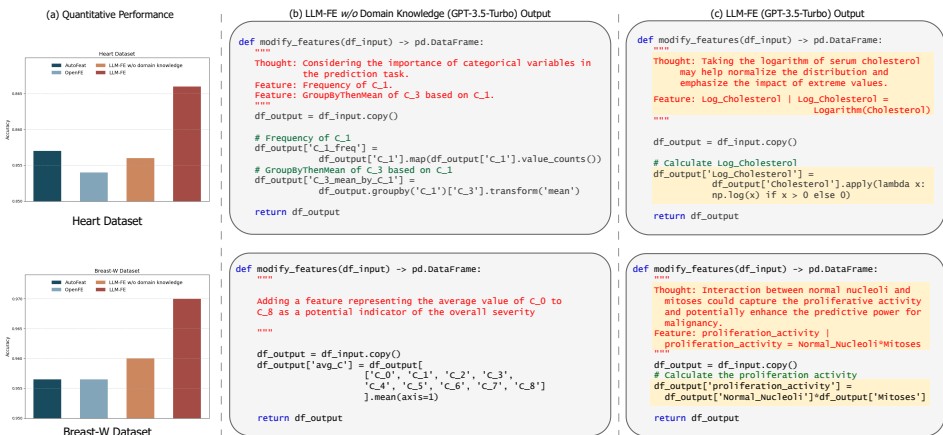

Figure 7: **Quantitative and Qualitative Analysis on Impact of Domain Knowledge for LLM-FE on Heart and Breast-W datasets.** (a) Comparison of XGBoost performance for LLM-FE against its domain-agnostic variant and traditional methods, such as OpenFE and AutoFeat, which do not integrate domain knowledge and exhibit reduced performance. (b) Features generated using the *w/o* Domain Knowledge variant of LLM-FE. (c) Feature discovery program generated by LLM-FE. The generated programs emphasize how incorporating domain expertise leads to more interpretable features that improve model performance.

## B.2 IMPACT OF DOMAIN KNOWLEDGE

Figure 7 highlights the qualitative and quantitative benefits of domain-specific feature transforms. We demonstrate this using two datasets: the Breast-W dataset, which focuses on distinguishing between benign and malignant tumors, and the Heart dataset, which predicts cardiovascular disease risk based on patient attributes. These tasks underscore the crucial role of domain knowledge in identifying meaningful features. Using embedded domain knowledge, LLM-FE not only significantly improves accuracy but also provides the reasoning for choosing the given feature, leading to more interpretable feature engineering. For example, in the Heart dataset, LLM-FE suggests the feature 'Log_Cholesterol', recognizing cholesterol's critical role in heart health and applying a logarithmic transformation to reduce the impact of outliers and stabilize the variance. In contrast, the '*w/o* Domain Knowledge' variant arbitrarily combines existing features, leading to uninterpretable transformations and reduced overall performance (Figure 7(a)). Similarly, for breast cancer prediction, LLM-FE identifies 'proliferation_activity' a biologically relevant metric leading to performance improvement, whereas the absence of domain knowledge results in a simple mean of all features, lacking interpretability and clinical significance (Figures 7(b) and 7(c)).

## B.3 IMPACT OF EVOLUTIONARY REFINEMENT

Figure 8 shows the detailed performance trajectory of LLM-FE compared with its '*w/o* Evolutionary Refinement' variant on PC1 and Balance-Scale datasets. The graph demonstrates that LLM-FE, using evolutionary search, consistently improves validation accuracy, while the non-refinement variant stagnates due to local optima. On the PC1 dataset, the non-refinement variant plateaus after seven iterations, and on the Balance-Scale dataset, it stagnates after five iterations. LLM-FE's evolutionary refinement helps it escape local optima with more robust optimization, leading to better validation accuracy on both datasets.

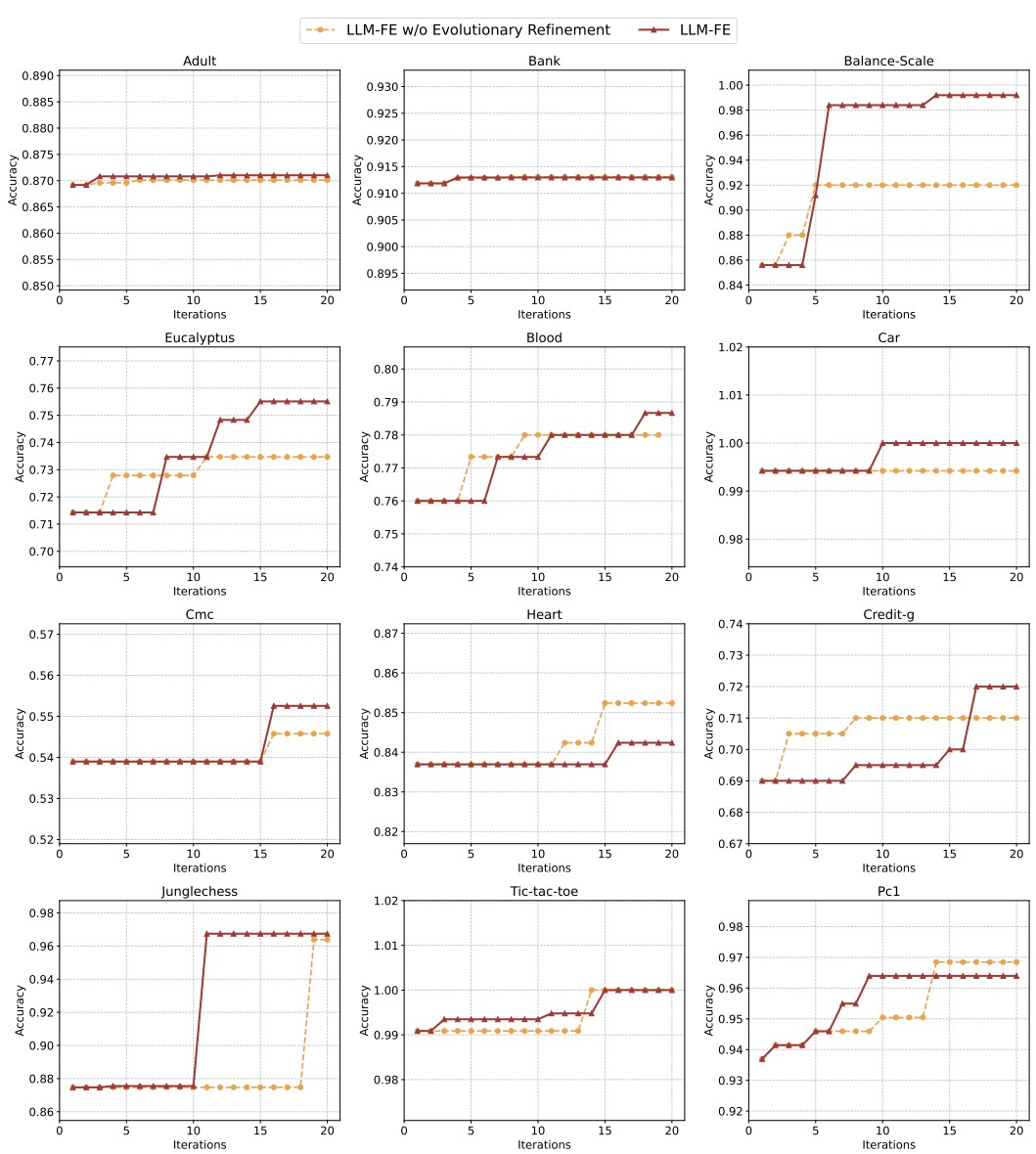

Figure 8: **Performance Trajectory Analysis.** Validation Accuracy progression for LLM-FE *w/o* evolutionary refinement and LLM-FE. LLM-FE demonstrates better validation accuracy, highlighting the advantage of evolutionary iterative refinement.

### B.4 Computational Analysis

Automated feature engineering methods, both classical and LLM-based, universally employ model training and validation to evaluate feature relevance. This evaluation strategy represents standard methodology across all automated feature engineering approaches rather than an additional computational burden specific to LLM-FE. We conduct our efficiency-performance trade-off analysis on the datasets with higher sample counts from Section 4.4, as these datasets present greater complexity with their substantial number of samples and features. Our comparative Pareto analysis (Figure 9) presents the base model alongside various feature engineering baselines. Our proposed method, LLM-FE, demonstrates Pareto optimality by achieving superior performance with substantially reduced computational requirements compared to existing methods, which either exhibit longer execution times or yield inferior performance metrics on these datasets. Only the base method requires less computation time than LLM-FE, but at a significant performance cost.

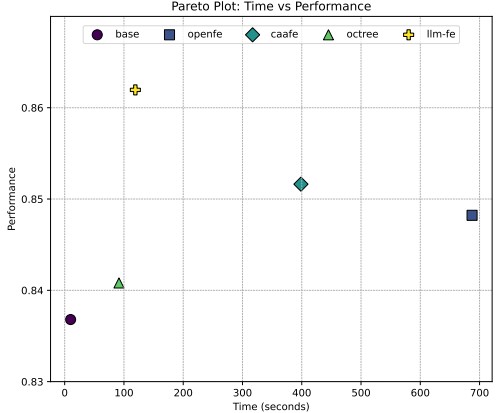

Figure 9: **Pareto Plot:** comparing trade-off between performance (accuracy) vs time (in seconds) for LLM-FE and other feature engineering baselines.

This positions LLM-FE as the optimal solution in the efficiency-performance space, delivering state-of-the-art results with reasonable computational demands even when handling datasets of considerable complexity.

### B.5 Memorization in Feature Engineering

Recent work has shown that LLMs can memorize tabular data under certain conditions (Bordt et al., 2024), motivating careful evaluation of their behavior on structured tasks. We therefore study XGBoost with and without LLM-FE, using `GPT-3.5-Turbo`, on datasets introduced by Bordt et al. (2024), which are specifically designed to test memorization and shown not to be present in model pretraining. In addition, we consider datasets from Hollmann et al. (2024), released after the September 2021 training cutoff for GPT models and hosted on Kaggle with hidden splits, making it unlikely they were included in pretraining corpora. As reported in Table 12, LLM-FE produces modest but consistent improvements across all datasets, indicating that the observed gains stem from semantically meaningful refinements rather than verbatim recall. While these findings highlight the value of domain-informed and evolutionarily refined features, memorization remains a critical concern in tabular domains, underscoring the need for more novel and carefully curated benchmarks.

Table 12: Comparison of the XGBoost with and without LLM-FE on five classification datasets.

| Dataset | Base | LLM-FE |
|---|---|---|
| kidney-stones | **0.761 ± 0.024** | **0.761 ± 0.027** |
| health-insurance | 0.756 ± 0.001 | **0.759 ± 0.001** |
| pharyngitis | 0.655 ± 0.008 | **0.660 ± 0.023** |
| fico | 0.715 ± 0.006 | **0.719 ± 0.009** |
| acs-income | 0.807 ± 0.002 | **0.809 ± 0.003** |

### B.6 LLM Bias in Operator Selection

LLMs exhibit a known bias toward simple operators such as addition, subtraction, or absolute value Küken et al. (2024), limiting the diversity of generated features relative to specialized feature-engineering systems. This tendency arises from pre-training corpora, where simple transformations dominate, making them 'default' choices even when more sophisticated transformations may be

beneficial. Despite this bias toward simpler operators, LLM-FE consistently demonstrates the valuable capability of identifying certain complex and informative transformations that are rarely generated by conventional LLM-based automated methods. Specifically, complex operators like `groupbythenmean`, `groupbythenmin`, `groupbythenmax`, `residual`, and `sigmoid` are also recommended frequently by LLM-FE, as illustrated in Figure 10. Such complex operations have the potential to capture meaningful patterns involving group-based aggregation that simpler transformations may miss. Thus, while further refinement is needed to balance operator selection, the ability of LLM-FE to discover nuanced, aggregation-based features emphasizes its promising role as a complementary technique in the broader automated feature engineering toolkit.

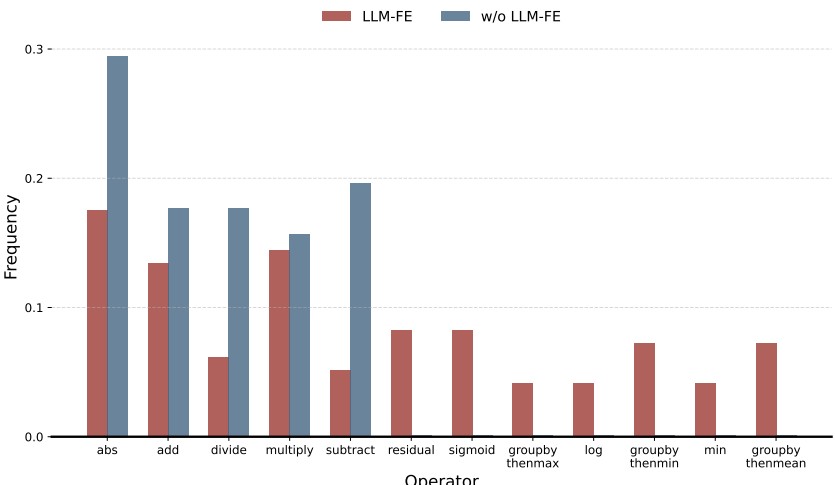

Figure 10: **Frequency of Feature Engineering Operators.** We compare the operators for LLM-FE with simple LLM-based methods.

## C  DATASET DETAILS

Table 13 describes the diverse collection of datasets spanning three major categories: (1) binary classification, (2) multi-class classification, and (3) regression problems used in our evaluation. The datasets were primarily sourced from established platforms, including OpenML Vanschoren et al. (2014); Feurer et al. (2021), UCI Asuncion et al. (2007), and Kaggle. We specifically selected datasets with descriptive feature names, excluding those with merely numerical identifiers. Each dataset includes a task description, enhancing contextual understanding for users. Our selection encompasses not only small datasets but also larger ones, featuring extensive data samples and high-dimensional datasets with over 50 features. This diverse and comprehensive selection of datasets represents a broad spectrum of real-world scenarios, varying in both feature dimensionality and sample size, thereby providing a robust framework for evaluating feature engineering works.

Table 13: Dataset statistics.

| Dataset | #Features | #Samples | Source | ID/Name |
|---|---|---|---|---|
| Binary Classification | | | | |
| adult | 14 | 48842 | OpenML | 1590 |
| blood-transfusion | 4 | 748 | OpenML | 1464 |
| bank-marketing | 16 | 45211 | OpenML | 1461 |
| breast-w | 9 | 699 | OpenML | 15 |
| credit-g | 20 | 1000 | OpenML | 31 |
| tic-tac-toe | 9 | 958 | OpenML | 50 |
| pc1 | 21 | 1109 | OpenML | 1068 |
| pima-indian-diabetes | 8 | 768 | OpenML | 43582 |
| Multi-class Classification | | | | |
| arrhythmia | 279 | 452 | OpenML | 5 |
| balance-scale | 4 | 625 | OpenML | 11 |
| car | 6 | 1728 | OpenML | 40975 |
| cmc | 9 | 1473 | OpenML | 23 |
| eucalyptus | 19 | 736 | OpenML | 188 |
| jungle_chess | 6 | 44819 | OpenML | 41027 |
| vehicle | 18 | 846 | OpenML | 54 |
| cdc diabetes | 21 | 253680 | Kaggle | diabetes-health-indicators-dataset |
| heart | 11 | 918 | Kaggle | heart-failure-prediction |
| communities | 103 | 1994 | UCI | communities-and-crime |
| myocardial | 111 | 1700 | UCI | myocardial-infarction-complications |
| Regression | | | | |
| airfoil_self_noise | 6 | 1503 | OpenML | 44957 |
| cpu_small | 12 | 8192 | OpenML | 562 |
| diamonds | 9 | 53940 | OpenML | 42225 |
| plasma_retinol | 13 | 315 | OpenML | 511 |
| forest-fires | 13 | 517 | OpenML | 42363 |
| housing | 9 | 20640 | OpenML | 43996 |
| crab | 8 | 3893 | Kaggle | crab-age-prediction |
| insurance | 7 | 1338 | Kaggle | us-health-insurancedataset |
| bike | 11 | 17389 | UCI | bike-sharing-dataset |
| wine | 10 | 4898 | UCI | wine-quality |

# D IMPLEMENTATION DETAILS

## D.1 BASELINES

We implement and evaluate various state-of-the-art feature engineering baselines, spanning traditional methods to recent LLM-based approaches, for comparison with LLM-FE. After generating features with each baseline, we apply a unified preprocessing pipeline to prepare the data for training and evaluation in the machine learning model. We implement the following baselines:

**AutoFeat.** AutoFeat is a classical feature engineering approach that uses iterative feature sub-sampling with beam search to select informative features. We utilize the open-source `autofeat`[1] package, retaining the default parameter settings. For parameter settings, we refer to the example '.ipynb' files provided in their official repository.

**OpenFE.** OpenFE is another state-of-the-art traditional feature engineering method using feature boosting and pruning algorithms. We employ the open-source `openfe`[2] package with standard parameter settings.

**FeatLLM.** FeatLLM uses an LLM to generate rules to binarize features that are then used as input to a simple model, such as linear regression. We adapt the open-source `featllm`[3] implementation, modifying the pipeline to use an `XGBoost` model for inference. To ensure a fair comparison with other methods, we provide the entire training dataset to train the `XGBoost` model while using only a subset of the dataset (10 samples) to the LLM to generate binary features. We report the results through an ensemble over three samples to maintain consistency with LLM-FE.

**CAAFE.** We utilize the official implementation of `CAAFE`,[4] maintaining all parameter settings as specified in the original repository. Following their workflow, we preprocess the data using their pipeline before inputting it into the prediction model after the feature engineering process.

**OCTree.** The official `OCTree` implementation[5] was modified to keep the data loading and model initialization part common. We implemented OCTree only for classification datasets, as the official implementation is limited to classification datasets, and running for regression datasets on our own could have resulted in incorrect implementation.

## D.2 LLM-FE

**Feature Generation.** Figure 12 presents an example prompt for the balance-scale dataset. The prompt begins with general instructions, followed by dataset-specific details, such as task descriptions, feature descriptions, and a subset of data instances serialized and expressed in natural language. To introduce diversity in prompting, we randomly sample between this approach and an alternative set of instructions, encouraging the LLM to explore a wider range of operators from OpenFE Zhang et al. (2023), as prior LLMs tend to favor simpler operators Küken et al. (2024) (see Figure 11). The quality of features generated has been detailed in Appendix B.6. By providing this structured context, the model can leverage its domain knowledge to generate semantically and contextually meaningful hypotheses for new feature optimization programs.

**Data-Driven Evaluation.** After prompting the LLM, we sample $b = 3$ outputs. Based on preliminary experiments, we set the temperature for LLM output generation to $t = 0.8$ to balance creativity (exploration) and adherence to problem constraints, as well as reliance on prior knowledge (exploitation). The data modification process is illustrated in Figure 12(c), where the outputs are used to modify the features via `modify_features(input)`. These modified features are then input into a prediction model, and the resulting validation score is calculated. To ensure efficiency,

---

[1]`https://github.com/cod3licious/autofeat.git`
[2]`https://github.com/IIIS-Li-Group/OpenFE.git`
[3]`https://github.com/Sungwon-Han/FeatLLM`
[4]`https://github.com/noahho/CAAFE`
[5]`https://github.com/jaehyun513/OCTree`

our evaluation is constrained by time and memory limits set at $T = 30$ seconds and $M = 2GB$, respectively. Programs exceeding these limits are disqualified and assigned None scores, ensuring timely progress and resource efficiency in the search process.

**Memory Management.** Following the 'islands' model used by Cranmer (2023); Shojaee et al. (2024); Romera-Paredes et al. (2024), we maintain the generated hypotheses along with their evaluation scores in a memory buffer comprising multiple islands ($m = 3$) that evolve independently. Each island is initialized with a basic feature transformation program specific to the dataset. Each island is initialized with a simple feature transformation program specific to the dataset (`def modify_features_v0()` in Figure 12(d)). In each iteration, novel hypotheses and their validation metrics are incorporated into their respective islands only if they exceed the island's current best score. Within each island, we additionally cluster feature discovery programs based on their signature, characterized by their validation score. Feature transformation programs that produce identical scores are consolidated together, creating distinct clusters. This clustering approach helps preserve diversity by ensuring that programs with varying performance characteristics remain in the population. We leverage this island model to construct prompts for the LLM. After an initial update of the prompt template with dataset-specific information, we integrate in-context demonstrations from the buffer. Following Shojaee et al. (2024); Romera-Paredes et al. (2024), we randomly select one of the $m$ available islands. Within the chosen island, we sample $k = 2$ programs to serve as in-context examples. To sample programs, we first select clusters based on their signatures using the Boltzmann selection strategy De La Maza & Tidor (1992) to sample clusters based on their signatures with a preference for clusters with higher scores. Let $s_i$ be the score of the i-th cluster, and the probability $P_i$ for selecting the i-th cluster is given as:

$$P_i = \frac{exp(\frac{s_i}{\tau_c})}{\sum_i(\frac{s_i}{\tau_c})}, \text{ where } \tau_c = T_0(1 - \frac{u \bmod N}{N}) \tag{4}$$

where $\tau_c$ is the temperature parameter, $u$ is the current number of programs on the island, and $T_0 = 0.1$ and $N = 10,000$ are hyperparameters. Once a cluster is selected, we sample the programs from it.

```
###
<Role>
You are a data scientist with expert knowledge about the provided dataset.
Your primary responsibility is to identify the most informative features that can enhance the solution to the
specified <Task>.

###
<Instructions>
  - You are given a task description, a list of existing features, a set of advanced operators, and sample
data.
  - Your objective is to leverage the provided advanced operators within <Operators> to generate meaningful
and insightful features that enhance task performance. These operators have been carefully curated to extract
deeper patterns from the data.
  - Avoid relying on basic arithmetic operators (e.g., addition, subtraction, multiplication, or division).
Instead, focus exclusively on the provided advanced operators inside <Operators>.
  - For each feature you derive, provide a concise explanation of why it is relevant and to solving the <Task>
in the docstring.

###
<Operators>
  - General Operators: Frequency (Frequency of feature in the data)
  - Numerical Input Operators: Absolute, Logarithm, Square Root, Sigmoid, Square, Round, Residual
  - Numeric-Numeric Operators: Minimum, Maximum
  - Categorical-Numeric Operators: GroupByThenMin, GroupByThenMax, GroupByThenMean, GroupByThenMedian,
GroupByThenStd, GroupByThenRank
  - Categorical-Categorical Operators: Combine, CombineThenFreq, GroupByThenNUnique
                                                                                        Instruction
```

Figure 11: **An example of the alternate set of instructions** used to direct the model to use a complex set of operations over simple operators for generating features.

```
###
<Role>
You are a data scientist expert in the field of the given dataset.
Your role is to apply your domain expertise to identify and create, and refine the most informative features
that solve the <Task> effectively.

###
<Instructions>
- You are provided with the task description, a list of existing features, and data examples.
- Use your domain knowledge to derive features that capture meaningful patterns, trends, or relationships
inherent in the data.
- Prioritize features that have high potential to enhance the model's ability to solve the <Task>, considering
both relevance and predictive power.
- For each derived feature, provide:
- A clear explanation of how it was derived and justification of its relevance for solving the <Task>.
- Ensure your approach remains grounded in the context of the dataset and the <Task>, and aim for features
that are interpretable and actionable.
```
**Instruction**

```
###
<Task>
Which direction does the balance scale tip to? Right, left, or balanced?

###
<Features>
- Left-Weight: Left-Weight (numerical variable within range [1, 5])
....
....

###
<Examples>
If Left-Weight is 3, Left-Distance is 3, Right-Weight is 4, Right-Distance is 5,  Then Result is right.
....
....

Please generate as many new features as possible using the information from the task, feature descriptions,
examples, and your domain understanding of the dataset. Remove any irrelevant, redundant, or less informative
features to enhance overall performance.

First, describe your new feature transformation and the main steps in a concise, one-sentence docstring.Then,
implement it in Python as a function that adheres to the given specifications.
Avoid providing any further explanations or additional descriptions.
```
**Dataset Specification**

```
def evaluate(data: dict):
    """ Evaluate the feature transformations on data observations."""
    import torch
    import utils
    from sklearn.model_selection import train_test_split
    from sklearn.metrics import accuracy_score
    from sklearn import preprocessing
    import xgboost as xgb

    #Data Loading and Processing

    # Load model
    model = xgb.XGBClassifier(random_state=42)
    # Training
    model.fit(X_train, y_train)
    # Inference
    y_pred = model.predict(X_test)
    score = accuracy_score(y_test, y_pred)

    return score, inputs, outputs
```
**Evaluation Function**

```
# Load data observations
label_encoder = preprocessing.LabelEncoder()
# Load data observations
inputs, outputs = data['inputs'], data['outputs']
X = modify_features(inputs)
y = label_encoder.fit_transform(outputs)
for col in X.columns:
    if X[col].dtype == 'string':
        X[col] = label_encoder.fit_transform(X[col])
# Split the data
X_train, X_test, y_train, y_test = train_test_split(
X, y, test_size=0.25, random_state=0)
# Data Processing
X_train = utils.make_numeric(X_train)
X_test = utils.make_numeric(X_test)

X_train = torch.tensor(X_train.to_numpy())
X_test = torch.tensor(X_test.to_numpy())
```

```
def modify_features_v0(df_input) -> pd.DataFrame:
    """
    Thought 1: The absolute difference between Left-Weight and Right-Weight can
    capture the imbalance in weight distribution.
    Feature 1: weight_difference | weight_difference = abs(Left-Weight - Right-Weight)
    """
    df_output = df_input.copy()
    # Calculate absolute difference between Left-Weight and Right-Weight
    df_output['weight_difference'] =
                abs(df_output['Left-Weight'] - df_output['Right-Weight'])

    return df_output
```
**In-Context Example**

```
def modify_features_v1(df_input) -> pd.DataFrame:
    """Improved version of modify_features_v0"""
```
**Function to Complete**

Figure 12: **Example of an input prompt for balance-scale dataset** containing (a) instruction, (b) dataset specification containing the details about the task, features, and data samples, (c) evaluation function, (d) initial in-context demonstration, and (e) function to complete.

