# OpenReview forum: "LLM-FE: Automated Feature Engineering for Tabular Data with LLMs as Evolutionary Optimizers"
_ICLR.cc/2026/Conference — ICLR 2026 Conference Withdrawn Submission_

### Official Review · Reviewer_pGeU · 2025-10-23

**Soundness:** 3
**Presentation:** 3
**Contribution:** 3
**Rating:** 4
**Confidence:** 5

**Summary:**

This paper introduces an LLM-based automated feature engineering method for tabular data. Specifically, they use the evolutionary algorithm. In  other words, LLM-FE guides the LLM to explore the code space by conditioning on the previous experiments. The experimental results show that LLM-FE outperforms previous SOTA baselines, e.g., OCTree, across various classification and regression tasks.

**Strengths:**

1. The authors considered various baselines including classical automated feature engineering methods and also LLM-based feature engineering methods.

2. Experimental results verify that LLM-FE is very effective across both classification and regression tasks.

3. Automated feature engineering is a challenging and important research direction for real-world application.

4. Paper is well-written and easy to follow.

**Weaknesses:**

1. My biggest concern is the Novelty of the paper. OCTree also leverages an evolutionary search algorithm guided by LLM to find a good feature engineering rule. Therefore, this paper just seems to be a naive expansion of OCTree. The only difference I think is that OCTree generates features one-by-one in a sequential manner, while LLM-FE do this at once.

2. Similary, I think LLM-FE is one of the applications of AlphaEvolve [1], which tries to refine the certain code block. As best as I know, LLM-FE also refines the code block to get a good feature generation function. In addition, this is also partially done by MLE-STAR [2], which explores feature engineering techniques by refining a corresponding code block.

[1] Novikov et al., AlphaEvolve: A coding agent for scientific and algorithmic discovery, 2025.

[2] Nam et al., MLE-STAR: Machine Learning Engineering Agent via Search and Targeted Refinement, 2025.

**Questions:**

See the above Weaknesses.

---

### Official Review · Reviewer_WjLx · 2025-10-29

**Soundness:** 2
**Presentation:** 3
**Contribution:** 2
**Rating:** 2
**Confidence:** 3

**Summary:**

In this paper, the authors propose a new method for feature engineering (referred to as FE below). In contrast to more traditional methods, which focus on selecting features based on a set of proposed rules, this method focuses on producing new features using prior knowledge contained by LLMs. This is not the first paper to propose utilizing LLMs for the task of FE in Tabular Machine Learning, however, as shown in Table 1, in contrast to other LLM-based approaches, such as CAAFE and OCTree, LLM-FE allows for arbitrarily complex features to be engineered and supports an approach that jointly generates and refines multiple features at once.

To achieve these positive qualities, the authors propose a combination of improvements to the process of FE.

First, authors include metadata available for a given dataset in the input prompt that an LLM receives. Additionally, authors explicitly instruct an LLM to generate complex features and provide step-by-step reasoning in favor of the relevance of the proposed features to the task at hand.

A second noticeable modification of the FE process includes providing data examples to an LLM, so that a model can see the interaction between the features and the target variable explicitly.

The next change is to produce new features directly via python code, as opposed to using some rigid structure of feature generation, such as decision trees, as was proposed in OCTree.

The final change that I would like to highlight in this review, is the use of evolutionary algorithm, that allows the model to refine the proposed features over many steps, using previously proposed sets of features as context in the LLM prompt.

The short summary of my opinion of the paper is provided below:

The paper is quite strong and novel, yet all the proposed changes are intuitively understandable. I believe that this paper ultimately steers the field of LLM-assisted feature engineering in the right direction, however there are some critical questions regarding evaluation, that make me recommend the rejection of the paper. With these questions addressed, I am very open to improving my score. I will summarize strengths and weaknesses of the paper in the corresponding sections below.

**Strengths:**

1. The paper is very easy to follow, with all the changes being introduced in order that makes it easy to understand them.
2. Using python code for feature generation removes any limitations from the process of feature engineering, so that any new feature can be hypothetically constructed with good enough LLM. This decision is very good for the overall state of the field, allowing for bigger utility from LLM assisted FE as the field progresses.
3. The ablation study is quite rigorous, showing that **each** of the novel components contributes to the improvement in metrics.
4. Resulting features can be interpreted by humans easily, which is important by itself in some fields and at the very least helps with the further feature engineering in others.

The paper is written well, steers the field in the right direction, and the proposed method is quite novel.

**Weaknesses:**

1. The main weakness that prevents me from giving a higher score, is the lack of statistical significance testing in the paper. Most increases shown in Table 2 and Table 3 seem to have very low statistical significance, but the paper does not mention this fact.
2. Hyperparameter optimization is a very important part of the pipeline for most tabular ML applications. However, it seems as only experiments from Table 5 in Appendix A.1 take it into account. Could you provide the results from Table 5 for all FE baselines and datasets?
3. On line 323, you write, “Finally, we sampled the top m (where m denotes the number of islands) feature discovery programs based on their respective validation scores and reported the final prediction through an ensemble”. Was ensembling also performed for all other baselines? Ensembling different models can increase the score on its own, so care is needed when evaluating ensembles of models.

Overall, my main concern is the statistical significance of the improvements from a proposed method. Perhaps, given small improvements provided by other methods, statistical significance from the proposed improvements is not necessary on most datasets, but should nevertheless be made clear in the paper. Also, importantly, hyperparameters should be optimized for all baselines and on all datasets, since HPO is a ubiquitous part of any tabular ML pipeline.

**Questions:**

1. Could you provide details on how many of the improvements are statistically significant?
2. In Table 5 in Appendix A.1, why only the “datasets where baseline models achieve accuracies below 0.8” were included? Why weren’t regression datasets included as well?
3. In Table 1, what do you refer to as “Complex Features”? Could you expand on that, and show that the features generated by your model are more complex than the ones generated by previous methods?
4. In Table 1, while other methods can’t refine multiple features at once, they can generate multiple new features consecutively. How many features do you generate with the baseline methods, and how many features does your model generate for each dataset?
5. On line 323, you mention that you evaluate an ensemble of models trained on different sets of features proposed by your model. How do you make this comparison fair to other baselines? I see that you mention an ensemble for FeatLLM in Appendix D.1, but what about other models? Also, do you select top-3 runs by validation score for them as well?

---

### Official Review · Reviewer_3dAZ · 2025-11-01

**Soundness:** 2
**Presentation:** 3
**Contribution:** 2
**Rating:** 4
**Confidence:** 4

**Summary:**

The paper proposes an LLM-based approach to automatic feature engineering in tabular learning, leveraging evolutionary search for more effective feature discovery than prior methods. While the use of LLMs as feature optimizers closely resembles several previous approaches, LLM-FE adopts a multi-population model to store and evolve features, promoting both diversity and effectiveness. Experimental results show that LLM-FE generates effective features for XGBoost, MLP, and TabPFN, outperforming both classical and LLM-based FE methods in terms of performance gains.

**Strengths:**

1. Clear overall framework. The proposed approach of leveraging LLM reasoning with an experience memory of candidate features is well-motivated and easy to follow.

2. Experiments with both classification and regression datasets with multiple predictors (XGBoost, MLP, TabPFN).

**Weaknesses:**

1. It is unclear which component of the framework is considered "evolutionary". While the approach is described as an evolutionary optimization method, it is not evident whether the feature optimization performed by the LLMs involves traditional evolutionary operations such as mutations or crossovers.

2. More discussion on computation cost is needed. LLM-FE evolves "islands" of programs which seems to make it potentially significantly more expensive than the baselines. More discussions on search efficiency, i.e., relative improvement in performance per number of candidate programs explored, would provide more valuable insights.

**Questions:**

Q. Have authors experimented with other types of LLMs, e.g., coding LLMs, which might make sense given that the task is generating features in Python programs?

Q. Do authors have explanations around why execution time exceeded 12 hours for some methods but not for LLM-FE? Given that LLM-FE manages multiple "islands" of programs, it is not clear conceptually whether LLM-FE has advantages over the baselines in terms of execution time.

Q. Does "specifically instruct(ing) the LLM to generate complex features" have any side effects (e.g., overfitting)?

---

### Official Review · Reviewer_TVLx · 2025-11-04

**Soundness:** 2
**Presentation:** 3
**Contribution:** 2
**Rating:** 4
**Confidence:** 4

**Summary:**

This paper introduces a framework LLM-FE leveraging LLMs as evolutionary optimizers to discover new features for tabular prediction. By combining LLM-driven hypothesis generation with data-driven feedback and evolutionary search, LLM-FE automates the feature engineering process. Through experiments on diverse tabular learning tasks, the authors demonstrate that LLM-FE outperforms state-of-the-art baselines and delivers improvements in predictive performance.

**Strengths:**

The writing of the paper is clear overall, making it easy to read. The proposed framework is evaluated across different tabular prediction tasks on real-world datasets and compared against the state-of-the-art FE baselines.

**Weaknesses:**

There seems to be limited improvement of the proposed framework from existing LLM-based FE approaches like CAAFE and OCTree apart from the sampling strategy of in-context examples. The design of the sampling strategy looks heuristic without detailed analysis.

Experimental evaluation is not complete. Table 3 does not include LLM-based FE baselines. LLM-FE is not compared against FE baselines for other downstream models in addition to XGBoost. The design choice of the sampling strategy requires further analysis. Parameter analysis of the framework is not provided.

Some experimental details are missing. The statistical tests and statistical significance are unclear. It is not reported how the hyperparameters of downstream models have been initialized. Ablation study and computational cost analysis do not contain much detail.

**Questions:**

What backbone LLM is evaluated in Tables 2 and 3?

---

### Note · Authors · 2025-11-28

I have read and agree with the venue's withdrawal policy on behalf of myself and my co-authors.